# Aromatic acids in an Arctic ice core from Svalbard: a proxy record of biomass burning

Mackenzie M. Grieman[1], Murat Aydin[1], Elisabeth Isaksson[2], Margit Schwikowski[3], and Eric S. Saltzman[1]

[1]Department of Earth System Science, University of California, Irvine, Irvine, California, 92697-3100, USA
[2]Norwegian Polar Institute Fram Centre NO-9296, Tromsø, Norway
[3]Paul Scherrer Institute, Villigen, Switzerland and Oeschger Centre for Climate Change Research, University of Bern, Switzerland

*Correspondence to:* Mackenzie M. Grieman (mgrieman@uci.edu)

**Abstract.** This study presents vanillic acid and para-hydroxybenzoic acid levels in an Arctic ice core from Lomonosovfonna, Svalbard covering the past 800 years. These aromatic acids are likely derived from lignin combustion in wildfires and long-range aerosol transport. Vanillic and para-hydroxybenzoic acid are present throughout the ice core, confirming that these compounds are preserved on millennial time scales. Vanillic and para-hydroxybenzoic acid concentrations in the Lomonosovfonna ice core ranged from below the limits of detection to 0.2 and 0.07 ppb, respectively (1 ppb = 1,000 ng L$^{-1}$). Vanillic acid levels are high (maximum of 0.1 ppb) from 1200-1400 CE, then gradually decline into the 20th century. The largest peak in the vanillic acid in the record occurs from 2000-2008 CE. In the para-hydrobenzoic acid record, there are three centennial-scale peaks around 1300, 1550, and 1650 CE superimposed on a long-term decline in the baseline levels throughout the record. 10-day air mass back trajectories for a decade of fire seasons (March-November, 2006-2015) indicate that Siberia and Europe are the principle modern source regions for wildfire emissions reaching the Lomonosovfonna site. The Lomonosovfonna data are similar to those from the Eurasian Arctic Akademii Nauk ice core during the early part of the record (1220-1400 CE), but the two ice cores diverge markedly after 1400 CE. This coincides with a shift in North Atlantic climate marked by a change of the North Atlantic Oscillation from a positive to a more negative state.

## 1 Introduction

Biomass burning influences the biosphere, atmospheric chemistry, and the climate system on both regional and global scales. Fire influences ecosystem dynamics, ecohydrology, surface albedo, and emissions of chemically and radiatively active aerosols and gases (Crutzen and Andreae, 1990; Legrand et al., 2016; Hessl, 2011; Bowman et al., 2009; Randerson et al., 2006). In boreal regions, fire plays a stabilizing role in circumboreal successional dynamics, influencing forest age structure, species composition, and floristic diversity (Soja et al., 2007). Boreal forest burned area, fire frequency, fire season length, and fire severity will likely increase with continued warming (Soja et al., 2007; Chapin et al., 2000). Arctic tundra fires are of particular concern because of their potential to release large amounts of ancient permafrost carbon into the atmosphere (Mack et al., 2011).

Understanding the role of fire in the climate system requires a knowledge of past regional/temporal variations on decadal, centennial, and millennial time scales. A number of proxy fire records have been developed from sediment cores and ice cores but systematic reconstruction of fire history remains a major challenge. Terrestrial sedimentary charcoal records are inherently local in extent, but regional and even global trends in burning have been developed from these records using various normaliz-
ing and averaging methods (Marlon et al., 2008, 2016; Power et al., 2008, 2013). The global charcoal database (GCD: Blarquez et al., 2014) is spatially and temporally inhomogeneous across the northern hemisphere boreal and Arctic regions, with good coverage in regions of North America and Western Europe, and poor coverage in Asia. Dissolved and particulate constituents in ice cores have also been used as burning proxies. These include cations (ammonium, potassium), anions (formate, acetate, nitrate), and black carbon (see Legrand et al., 2016; Rubino et al., 2016, for recent reviews). One of the challenges of interpret-
ing these records is that most of the dissolved ions have multiple sources, in addition to burning. For example, ammonium is also derived from biogenic marine and terrestrial sources, agriculture, and livestock (Legrand et al., 1992, 2016; Fuhrer et al., 1996; Savarino and Legrand, 1998; Bouwman et al., 1997; Rubino et al., 2016). Efforts to isolate the fire-derived contributions to these records have employed principle component analysis and peak counting methods. Eichler et al. (2009) examined a Siberian Altai ice core using a multiproxy approach, and concluded that potassium, nitrate, and charcoal were fire-related
while ammonium and formate were biogenic in origin. The detailed interpretation of ice core chemical proxies is complicated by the fact that black carbon is emitted primarily during the flaming conditions, while ammonium and many organic aerosol-borne constitutents are emitted primarily during smoldering. Ice core gas measurements of methane and carbon monoxide, and their isotopomers, have also been used to derive histories of pyrogenic emissions (Ferretti et al., 2005; Wang et al., 2010). These gases have sufficiently long atmospheric lifetimes that they integrate emissions over hemisphere/global scales.

A variety of organic aerosols are emitted from the burning of vegetation under smoldering conditions. Levoglucosan, a combustion product of cellulose, is considered a universal biomass burning tracer because it is emitted in greater quantities than most other burning-derived organic aerosols and is uniquely produced by the burning of plant matter (Simoneit et al., 1999). Levoglucosan has been detected in ice cores from Antarctica, Greenland, and northeast Asia (Gambaro et al., 2008; Kawamura et al., 2012; Zennaro et al., 2014). It is considered a qualitative tracer because it degrades rapidly in the atmosphere
(Hoffmann et al., 2010; Hennigan et al., 2010; Legrand et al., 2016; Slade and Knopf, 2013).

Aromatic acids are among a wide range of phenolic compounds generated by lignin pyrolysis. These compounds are ubiquitous constituents of biomass burning aerosols and have been detected in polar ice cores. Lignin is produced from three precursor alcohols (p-coumaryl alcohol, coniferyl alcohol, and sinapyl alcohol), and the resulting phenolic compounds retain the structure of these alcohols. The aromatic acids analyzed in this study are vanillic acid (VA) and p-hydroxybenzoic acid
(p-HBA). VA is predominantly associated with conifer and deciduous boreal forest tree species, while tundra grasses and peat generate primarily p-HBA with lesser amounts of VA (Simoneit, 2002; Oros and Simoneit, 2001a, b; Oros et al., 2006; Iinuma et al., 2007). p-HBA is also produced from conifer boreal tree species (Oros and Simoneit, 2001a).

Burning is the only known source of these aromatic acids in aerosols or ice cores. Quantitatively, the ice core levels of these compounds result from the combined effects of emissions, atmospheric transport, depositional, and perhaps post-depositional
processes. Aromatic acids can undergo re-volatilization at the snow surface. Laboratory experiments have shown that Arc-

tic snow samples containing lignin-derived compounds photochemically react to produce formaldehyde and acetaldehyde (Grannas et al., 2004). Melting and refreezing processes have the potential to redistribute aromatic acids to lower depths. Meltwater at the surface percolates to deeper snow layers and water-soluble compounds are concentrated when the meltwater is refrozen (Wendl et al., 2015).

5     Prior studies of pyrogenic aromatic acids in ice cores include shallow cores from Greenland, northeast Asia (Kamchatkan Peninsula), and Europe (the Swiss Alps) (McConnell et al., 2007; Kawamura et al., 2012; Müller-Tautges et al., 2016). The record from Greenland showed that the timing of variability in the VA and black carbon records was similar over the past 200 years until around 1890 CE (McConnell et al., 2007). VA and p-HBA were elevated from the 1950's to the 1970's in the 60-year ice core record from the Swiss Alps (Müller-Tautges et al., 2016). VA and p-HBA were elevated in the 1700's and in the 20th century in the ice core from Northeast Asia over the past 300 years. There is only one millennial time scale ice core record of VA and p-HBA: a 2600-year Akademii Nauk ice core record from the Eurasian Arctic (Grieman et al., 2017). That record showed three major multi-century pulses of burning-derived aromatic acids, including one during the Little Ice Age (1450-1700 CE).

     Here we present measurements of vanillic acid (VA) and p-hydroxybenzoic acid (p-HBA) in an Arctic ice core from the Lomonosovfonna ice cap in central Spitsbergen, Svalbard, which is located northeast of Greenland, in the Atlantic sector of the Arctic ocean (Fig. 1). The goal of this study was to generate a record sensitive to conditions in Northern Europe/Northern Eurasia. Air mass back trajectories are used to examine the distribution and ecology of likely source regions for biomass burning aerosols transported to Svalbard. We discuss the variability observed in the ice core records of these compounds over the past 750 years, and compare the records to other proxy records of northern hemisphere climate and biomass burning.

## 2   Methods

### 2.1   Ice core site characteristics and dating

The Lomonosovfonna ice core site is 1202 m above sea level (asl) (78.82°N, 17.43°E) (Fig. 1). The ice core was drilled in 2009 to a depth of 149.5 m by a team from the Paul Scherrer Institute and the Norwegian Polar Institute. The core did not reach bedrock and contains a continuous 750-year record (Wendl et al., 2015). The near-surface annual average temperature of Lomonosovfonna is -12.5°C at 1020 m asl (Beaudon et al., 2013; Pohjola et al., 2002; Zagorodnow, 1988). The annual average accumulation rate is $0.58 \pm 0.13$ m water equivalent (meq) year$^{-1}$ (Wendl et al., 2015). The firn/ice transition occurs at 13 m depth, at approximately 1997 C.E.

     The Lomonosovfonna ice core was dated by Wendl et al. (2015), yielding a time span for the core of 1222-2009 CE (Fig. S1). Annual layers were counted using seasonal $\delta^{18}$O and Na$^+$ variations to a depth of 97.7 m (79.7 m weq), giving an age of 1750 CE at that depth. The chronology of the upper section of the core was also constrained by the $^{210}$Pb profile and the 1963 $^3$H horizon. The age scale below 97.7 m was developed using a simple glacier ice flow model (Thompson et al., 1998), assuming an average accumulation rate of $0.58 \pm 0.13$ m weq year$^{-1}$ (Wendl et al., 2015). The age scale was adjusted to match seven volcanic reference horizons. The oldest of these is the Samalas volcanic eruption of 1257/8. Dating uncertainty

was estimated by comparing purely modelled reference horizon years to known volcanic eruption years. Above 68 m weq, the dating uncertainty is $\pm 1$ year within 10 years of reference horizons, and $\pm 3$ years otherwise. Between 68-80 m weq, the dating uncertainty is $\pm 3$ years, and below 80 m weq, the dating uncertainty is $\pm 10$ years (Wendl et al., 2015).

The Lomonosovfonna site experiences summer surface melting and winter refreezing (Wendl et al., 2015). Wendl et al. (2015) examined the distribution of melt layers in the Lomonosovfonna ice core, concluding that most summer melt layers are refrozen within the year, with some extending over 2-3 years. The frequency of melt layers increases after 1800 CE (Fig. S2; Wendl et al., 2015). During the warmest years of the twentieth century, percolation length reached 8 years. Due to possible redistribution of soluble compounds by melt, percolation, and refreezing, inter-annual variability of the aromatic acid signals is not interpreted in this study. Ten-year bin averages are used to illustrate short-term variability in the data (see Section 3.1).

## 2.2 Potential source regions and ecological types using air mass back trajectories

Air mass back trajectories were used to identify potential source regions and ecofloristic zones from which biomass burning aerosols are likely to reach the Lomonosovfonna ice core site. This analysis assumes modern-day meteorological conditions. 10-day air mass back trajectories were computed using the HYSPLIT model with NCEP/NCAR Reanalysis data (Draxler et al., 1999; Stein et al., 2015; Kalnay et al., 1996). The 10-day back trajectories were started at 100 m above the ice surface at 12:00 AM and 12:00 PM local time for three separate 10-year periods, 1948-1957, 1970-1979, and 2006-2015 CE. The fraction of trajectories originating in or transecting various geographic regions and ecofloristic zones was calculated for spring (March 1-May 31), summer (June 1-Aug. 31), and fall (Sept. 1-Nov. 30). The geographic regions included in the study were North America, Siberia ($>42°$E), and Europe ($<42°$E). The boundaries of North America, Siberia, and Europe were defined using global self-consistent, hierarchical, high-resolution geography database GIS shapefiles (Wessel and Smith, 1996). These regions were subdivided into ecofloristic zones defined by the Food and Agriculture Organization (FAO; Fig. S3; Ruesch and Gibbs, 2008, http://cdiac.ornl.gov/epubs/ndp/global_carbon/carbon_documentation.html).

The Siberian region has the highest fraction of trajectories to the Lomonosovfonna site, accounting for 39%, 15%, and 38% of the trajectories in spring, summer, and fall from 2006-2015 CE, respectively. Siberian trajectories were most commonly from boreal tundra woodlands, boreal conifer forests, and boreal mountain systems (Fig. S4; Table S1). Fewer than 3% of the trajectories transected other Siberian ecofloristic zones. Europe contributed fewer than 11% of the trajectories arriving at the site in any season. Those trajectories most commonly encountered boreal coniferous forests, boreal mountain systems, and temperate oceanic forests (Fig. S5; Table S1). Pollen species in the Lomonosovfonna ice core covering the past 150 years drilled in 1997 match northern boreal taxa from Fennoscandia (Hicks and Isaksson, 2006). Biomass burning aerosols from Eastern European agricultural fires in 2006 reached Svalbard within a few days (Stohl et al., 2007). Other European ecofloristic zones contributed to fewer than 5% of the trajectories in any season. North America contributed fewer than 4% of the trajectories for any season. This analysis does not rule out contributions from North America, but it does suggest that such input would likely require considerably longer atmospheric transport times.

## 2.3 Ice core sample preparation and analysis

For this study, we resampled discrete ice core samples previously analyzed for major ions (Wendl et al., 2015). The original ice core samples were 1.8x1.9 cm in cross section and 3-4 cm long, melted and stored frozen in polypropylene vials. For analysis of VA and p-HBA, the ice was re-melted, 1 ml was withdrawn from each vial, and the samples from four adjacent vials were combined into one. This resulted in a total of 993 samples. As discussed below, this sampling procedure resulted in decreasing temporal resolution with increasing depth from subannual samples at the surface to about 2-year samples at the bottom of the core.

VA and p-HBA in the ice core samples were measured using anion exchange chromatographic separation and tandem mass spectrometric detection in negative ion mode with an electrospray ionization source (IC-ESI-MS/MS). The analytical methods and standards used in this study are described in detail by Grieman et al. (2017). The experimental system consisted of a Dionex AS-AP autosampler, ICS-2100 integrated reagent-free ion chromatograph, and ThermoFinnigan TSQ Quantum triple quadrupole mass spectrometer. VA was detected at two mass transitions (m/z 167→108 and m/z 167→152) and p-HBA was detected at m/z 137→93.

Limits of detection for single measurements were estimated using three times the standard deviation of the MilliQ water blank. The limits of detection for the VA m/z 167→108 and 167→152 transitions were 0.010 and 0.006 ppb (n = 80), respectively. The limit of detection for p-HBA was 0.012 ppb (n = 80). The mass spectrometer signals for VA at the two mass transitions were highly correlated and either mass transition can be used to measure ice core VA (Fig. S6). Data from the m/z 167→152 transition is reported here because of the slightly better detection limit.

## 3 Results and Discussion

### 3.1 Analytical results and data processing

In this study, 993 samples were analysed for VA and p-HBA (Fig. 2). VA and p-HBA levels range from below detection to 0.2 and 0.07 ppb, respectively. A substantial fraction of the VA and p-HBA data (67% and 58%, respectively) is below the limits of detection. Data below the limits of detection are reported as 0.5 times the limit of detection (0.003 ppb for VA and 0.006 ppb for p-HBA). Smoothing of the data was carried out using time bin averaging (10, 40, and 100-year), LOESS smoothing, and moving averages. All smoothing treatments reveal similar multi-decadal and centennial-scale features in the records and the choice of smoothing technique does not influence the interpretation of the data (Fig. S7; Fig S8). Geometric means and standard deviations of the transformed data were used for all statistics because frequency distributions of the data show skewness towards lower concentrations. Time-averaging compensates for the decrease in frequency of sampling with depth in the core due to layer thinning.

## 3.2 The Lomonosovfonna vanillic and p-hydroxybenzoic acid time series

The Lomonosovfonna VA and p-HBA time series exhibit variability on a wide range of time scales. There is abundant annual to decadal variability in both records. The amplitude of individual peaks in the raw data is roughly similar across the whole record for both compounds, ranging from 0.1-1.2 ppb for VA and 0.1-0.8 for p-HBA (Fig. 2). The peaks appear to be of longer duration during the older portions of both records. Both of these aspects of the raw data are likely artifacts due to the combined effect of annual layer thinning with depth and the sampling strategy of analyzing individual ice core samples of constant thickness (12-16 cm). The time span integrated by individual samples ranged from 1.7 years near the bottom of the core, to 0.5 years at mid-depth ($\sim$80 m), to 0.07 years near the top (Fig. S1). This thinning effect is eliminated when the data are time bin-averaged. Peaks in the 10-year bin-averaged records are roughly similar in duration across the whole record (Fig. 3). In the bin-averaged data, the magnitudes of the peaks are no longer constant across the record. For VA, the two early peaks (1250-1280 and 1360-1390 CE) are much larger than all subsequent peaks. p-HBA exhibits three major multi-decadal peaks. One is simultaneous with the early VA peak (1250-1280 CE) and the others occur from 1520-1570 and 1610-1640 CE.

Both compounds exhibit long-term decreasing trends over the 800-year record, as illustrated by the 40-year bin averaged data (Fig 3). 40-year bin-averaged VA levels decline by about a factor of three over the first half of the record (1200-1600 CE), then remain relatively steady for the remainder of the record. It is possible that the decline in VA continued after 1600 CE but much of the data after this time is near or below detection. 40-year bin-averaged p-HBA levels decline by about a factor of two over the whole 800-year record.

Centennial scale variability is observed as pronounced maxima early in the record (1300-1500 CE), as illustrated by the 40-year bin averages (Fig. 3). There are hints of continued centennial scale variability in VA in the remainder of the record. Centennial scale variability is evident throughout the p-HBA record, with maxima coinciding with the VA maxima early on and with additional maxima in the 1500's, 1600's, and 1800's.

The 20th century levels of VA and p-HBA are not anomalous relative to the rest of the ice core record. VA exhibits a slight increase after 1970 and the largest single peak in the record occurs from 2000-2008 CE (Fig. 4). p-HBA levels also appear to increase after 1970, although to a lesser degree than VA. The 2000-2008 period is slightly elevated in p-HBA but not to the extent observed in VA. The samples from 1997-2009 CE are within the firn layer. It is possible that firn samples could be contaminated with biomass burning aerosols during handling in the field but we have no reason to suspect that the aromatic acids in these samples are influenced by contamination. We have not observed laboratory contamination for these compounds as a significant issue in our laboratory.

Wavelet analysis was used to illustrate temporal variations in the spectral content of the signals. Lomonosovfonna VA and p-HBA time series exhibit non-stationary periodic variability, meaning that the spectral characteristics vary with time (Fig. S9).

## 3.3 Potential for post depositional modification of VA and p-HBA

There have been no field studies of atmosphere/snow interactions for aromatic acids like VA and p-HBA, so little is known about postdepositional effects. Three types of effects should be considered: 1) revolatilization after deposition to the snowpack,

2) vertical redistribution associated with melting, percolation, and refreezing, and 3) degradation due to chemical or microbiological processes. All of these effects would likely occur to a greater extent at relatively warm sites like Lomonosovfonna (mean annual temperature -12.5°C), and during warmer periods like the Medieval Climate Anomaly or the 20th century. Redistribution associated with melt layers has been discussed in detail for other ions (Wendl et al., 2015), and one would expect that the influence of these processes on aromatic acids would be similar. Wendl et al. (2015) used principal component analysis to determine that melt layers did not have a major influence on the distribution of ions on decadal timescales. Finally, the VA and p-HBA data from Lomonosovfonna and Akademii Nauk argue against chemical degradation as an important process, since there is clearly no monotonic decrease in VA or p-HBA levels downcore.

The fact that VA and p-HBA are commonly observed in atmospheric aerosols, even after long distance transport, suggests that the volatility of these compounds in aerosols might be considerably lower than that of the pure substance (Simoneit and Elias, 2000; Simoneit et al., 2004; Zangrando et al., 2013, 2016). The vapor pressures for VA and p-HBA are 0.0023 pa and 2.5 x $10^{-5}$ pa (https://chem.nlm.nih.gov/chemidplus/rn/121-34-6 Jones, 1960) at 25°C, respectively. Ionic interactions with salts or hydrophobic interactions with soot or complex organics may stabilize aromatic acids in aerosols or snow. Laboratory and aerosol field studies have demonstrated reduced volatility of low molecular weight organic acids (relative to the vapor pressure of the pure compound) due to interaction with cations derived from seasalt or other sources, but this effect has not been studied for aromatic acids (Häkkinen et al., 2014; Laskin et al., 2012).

If revolatilization of aromatic acids from the snowpack does occur, one might expect loss to be enhanced in ice acidified by high levels of nitrate and sulfate from volcanic or pollutant inputs. There is no obvious evidence that acidification is a dominant control on VA or p-HBA levels in the ice core (Fig. S10). It is particularly notable that VA and p-HBA levels are not anomalously low during the twentieth century, when ice core sulfate and nitrate levels increased several-fold compared to the preindustrial era (Fig. S10; Fig. S11). Based on the ice core signals alone, we conclude that re-volatilization does not appear to be the predominant factor controlling ice core aromatic acid levels, although this cannot be ruled out. Further investigation of this issue is needed.

### 3.4   Relationship to ammonium record

Here we compare the variability of Lomonosovfonna VA and p-HBA to the previously published ammonium record from the same ice core (Wendl et al., 2015). That study concluded that prior biogenic sources were the major contributor to ammonium in the ice core prior to the mid-1800's, and agriculture became a major source during the 20th century. Prior studies have suggested that episodic ammonium peaks in ice cores represent a fire signal, while longer-term variability reflects the biogenic signal (Fischer et al., 2015; Legrand et al., 2016). Following Fischer et al. (2015), we used singular spectrum analysis (SSA) to decompose the Lomonosovfonna VA, p-HBA, and ammonium records into these two components.

The analysis was done by computing 30 principle components (PC's) using 3-year bin averaged data for VA, p-HBA, and ammonium (Wendl et al., 2015). The low frequency component (PC-1) of the ammonium record shows little similarity to PC-1 for either of the organic acids. VA and p-HBA exhibit decreasing trends over the record while ammonium increases (Fig. 5). To compare the higher frequency components, we used a peak detection method (Higuera et al., 2010). This was done by summing

PC's 2-30 and counting the frequency of peaks above a threshold (75th percentile) in a 40-year moving window. The resulting signals for VA and p-HBA exhibit centennial-scale variability that is consistent with that obtained from bin-averaging (Fig. 3) and ammonium exhibits similar variability on these time scales. The correlation coefficient between VA-ammonium and p-HBA-ammonium was computed from the peak frequency data using a 200-year moving window. The 95% confidence interval
for these correlation coefficients are shown in Fig. 5. Based on this analysis, ammonium and VA are positively correlated for three time periods (1300-1450; 1675-1725; post-1880). Ammonium and p-HBA exhibit positive correlations for two time periods (1425-1650; 1825-1875). Interestingly, the positive correlations for ammonium with VA and with p-HBA occur at different intervals. The fact that some extended periods of correlation between VA, p-HBA, and ammonium are present in the Lomonosovfonna record suggests that there may be a fire-derived contribution to the ammonium signal in this ice core.
However, the relationships are obviously complex and worthy of further study.

## 3.5 Relationship to sedimentary charcoal records

Sedimentary charcoal records in the Global Charcoal Database (GCD) from Siberia (50-70°N, 50-150°E) and Fennoscandia (50-70°N, 0-50°E) were analyzed using the paleofire R package (GCD: Blarquez et al., 2014). Only 3 of the 12 Siberian records in the GCD have sufficient data from 1200-2000 CE for comparison to the Lomonosvfonna ice core record. These regions are
Chai-ku Lake in eastern Siberia, and Zagas Nuur and Lake Teletskoye in southern Siberia. The Siberian region as a whole is therefore not well-represented. Six of the 19 Fennoscandian records in the GCD have enough data from 1200-2000 CE for comparison to the Lomonosovfonna ice core record. Siberia and Fennoscandia are primarily boreal tundra woodlands, boreal conifer forests, and boreal mountain systems (Fig. S3; Ruesch and Gibbs, 2008). One important caveat to this comparison is that the dating of sedimentary charcoal records is often based on linear interpolations between a few [14]C ages. Hence their age
scales are typically less well-constrained than Lomonosovfonna or other ice cores.

Four of the six records from Fennoscandia exhibit increased charcoal influx from 1200-1400 CE (Fig. S12; Blarquez et al., 2014). Lomosovfonna VA and p-HBA are both elevated during this period. Three of the six charcoal records are elevated around 1600 CE when Lomonosovfonna p-HBA is also elevated. Two of the records also show a long-term decline from 1200-2000 CE similar to the Lomosovfonna VA and p-HBA records. Two of the Siberian records exhibited increased charcoal influx from 1200-1600 CE relative to 1600 CE to present (Fig. S13). The Lomonosovfonna VA and p-HBA are also higher early in
the record. The Fennoscandian records are clearly most similar to the Svalbard ice core record, but the database is too limited to determine definitively the srouce region for the VA and the p-HBA in the Lomonosovfonna ice core from so few charcoal records.

## 3.6 Comparison between Svalbard and Siberian ice core records of vanillic acid and p-hydroxybenzoic acid

The only other millennial scale ice core record of VA and p-HBA is the Akademii Nauk ice core from the Severnaya Zemlya Archipelago in the Arctic Ocean north of central Siberia (Grieman et al., 2017; Fritzsche et al., 2002). The Akademii Nauk ice core covers a considerably larger time range than Lomonsovfonna, extending over the past 2600 years. Here we discuss only the period of temporal overlap between the two ice cores (1200-2000 CE).

The two ice core records exhibit similar trends and levels during the early part of the record (1220-1400 CE; Fig. S14). During this period, Lomonosovfonna exhibits declining levels of both aromatic acids. In the Akademii Nauk core, this period represents the tailing end of an earlier peak in both aromatic acids with a maximum around 1200 CE. The two cores diverge markedly after 1400 CE for the remainder of the records (Fig. 6). Akademii Nauk VA exhibits a period of highly elevated levels from 1460-1660 CE. During this period, Akademii Nauk VA reaches levels more than an order of magnitude above those of Lomonosovfonna VA. Akademii Nauk p-HBA exhibits a period of elevated levels from 1460-1550 and 1780-1860 CE. There are multi-decadal peaks in p-HBA in the Lomonosovfonna record that overlap in time with the large Akademii Nauk VA peaks, although not nearly as large in amplitude or duration. Interestingly, these peaks do not appear at all in the Lomonosovfonna VA record.

10-day back trajectories were computed for the Akademii Nauk site using the same methods as those described above from 2006-2015 CE (section 2.2; Grieman et al., 2017). The trajectories show that both of the Lomonosovfonna and Akademii Nauk sites are influenced by air masses transecting Eurasian forested regions (Fig. 7; Table S1). The largest fraction of trajectories transect Siberian boreal tundra woodland, boreal coniferous forests, and boreal mountain systems for both ice core sites, particularly in the summer and fall. Given this similar transport pattern, we would have expected the two large multi-century peaks in Siberian aromatic acids after 1400 CE to be exhibited in the Lomonosovfonna record as well. The sharp divergence between the two records around 1400 CE and the subsequent dramatic increase in aromatic acids only in the Siberian ice core suggest a change in transport patterns to the sites after 1400 CE. The fraction of back trajectories transecting vegetated regions of Siberia for Akademii Nauk were about twice that for Lomonosovfonna. Conversely, trajectories from European forests comprise a smaller contribution to Akademii Nauk than to Lomonosovfonna. Air masses from European regions are more likely to reach the Lomonsovfonna site than the Akademii Nauk site. We speculate that the divergence between the two ice cores reflects a shift in large-scale atmospheric circulation patterns, as discussed below.

## 3.7 Relationship to atmospheric circulation and climate

The general climate context for the last millennium is Late Holocene cooling, with superimposed centennial scale climate variability associated with the Medieval Climate Anomaly (MCA, 950-1250 CE), the Little Ice Age (LIA, 1400-1700 CE), and the twentieth century warming (PAGES 2k Consortium, 2013; Lamb, 1965; Mann et al., 2009). Svalbard $\delta^{18}$O ice core records show that cooling continued in the region through the 19th century (Divine et al., 2011). Divine et al. (2011) suggest that the extended LIA at Svalbard could reflect the climatic influence of regional sea ice conditions. In that case, the extended LIA was probably not characteristic of the biomass burning source regions in Europe and Siberia influencing the Lomonosovfonna ice core.

For recent decades, increased burning of wildfires is generally associated with higher summer temperatures (Flannigan et al., 2009). On that basis alone, one might expect to see a long term decrease in aromatic acid signals over the last millennium, and that is generally the case for both VA and p-HBA in the Lomonosovfonna ice core. However, on multi-century and centennial time scales, the variability in the aromatic acids in the Lomonosovfonna ice core is also large and somewhat complex. Both VA and p-HBA levels were high during the MCA. VA declines into the LIA. p-HBA exhibits elevated levels during the latter

half of the LIA but VA does not (Fig. 8). This dissimilarity could be due to a shift in spatial patterns of either biomass burning or atmospheric transport after 1400 CE.

It seems likely that regional changes in burning proxies on multi-century and centennial time scales are strongly linked to changes in large scale atmospheric circulation and the resulting impacts on regional climate and atmospheric transport. For the source regions influencing the Svalbard Lomonosovfonna ice core, one might expect that changes in the North Atlantic Oscillation (NAO) might play an important role. The NAO is a major mode of climate variability in the North Atlantic region, characterized by changes in the pressure gradient between the Icelandic Low and the Azores High during winter months (Hurrell et al., 2001). Strong pressure gradients (positive NAO index) are associated with strong zonal flow, enhanced westerlies transporting warm air to Europe, increased precipitation in northwest Europe, and decreased precipitation in southern Europe (Trouet et al., 2009). Weaker pressure gradients (negative NAO index) are associated with stronger meridional flow and cooling of the North Atlantic region (Trouet et al., 2012). Proxy NAO records have been developed from variations in wintertime seasalt sodium in the GISP2 ice core, from Moroccan tree-rings and speleothem records in Scotland, and from lake sediments in southwestern Greenland (Meeker and Mayewski, 2002; Trouet et al., 2009; Olsen et al., 2012).

The proxy NAO records show a marked change in phase at the onset of the LIA (around 1400 CE) from several hundred years of positive NAO index to a more negative and variable NAO state that continued throughout and after the LIA (Fig. 8). The Lomonosovfonna oxygen isotope ($\delta^{18}$O) record shows a cooling trend at this time, consistent with the NAO shift (Wendl et al., 2015). The change in NAO behaviour at this time was accompanied by a decline in VA in the Lomonosovfonna record, a decline in the VA/HBA ratio, and a sudden divergence between the Lomonosovfonna and Akademii Nauk ice cores (1400 CE). We suggest that a change of high latitude northern hemisphere atmospheric circulation patterns occurred at this time, resulting in 1) a cooler, wetter northern Europe with less burning, and 2) reduced zonal transport, resulting in 'decoupling' of the two ice core signals. Central Siberian burning likely increased at this time, as evidenced by sharply increased aromatic acids in the Akademii Nauk ice core. The Siberian ice core signals are similar in timing to changes in the strength of the Asian monsoon, as recorded in speleothem proxy records (Grieman et al., 2017; Wang et al., 2005). We speculate that during the LIA, central Siberia was influenced primarily by conditions in the Pacific rather than the Atlantic Ocean.

The summertime NAO (SNAO) is defined as the leading mode of July–August sea level pressure variability in the North Atlantic sector (Folland et al., 2009; Efthymiadis et al., 2011). The SNAO affects temperatures, precipitation, and cloudiness in Europe during summer, and one might expect that variations in burning are even more directly linked to the SNAO than the NAO. The SNAO has a slightly different spatial pattern than the NAO, with a relatively small Arctic node and a southern node over northwestern Europe. The positive (negative) mode of the SNAO is characterized by a warmer and drier (cooler, wetter) northern Europe (Linderholm et al., 2008; Folland et al., 2009). The influence of the SNAO extends to central Asia, and so could influence both major source regions for the Lomonosovfonna ice core.

In order to illustrate the possible influence of the SNAO, we compared back trajectories from the Lomonosovfonna site for recent periods when the SNAO index was positive (1970-1979 CE, mean SNAO index: 6.3) and negative (1948-1957 CE; mean SNAO index: -2.0) (Folland et al., 2009). Figure 9 shows the major spatial clusters of 10-day air mass back trajectories for each time period (computed using Hysplit) superimposed on the sea level pressure (SLP) anomalies relative to mean SLP from

1948-2017. The high SNAO period is characterized by 1) high pressure over Scandinavia, favoring drier conditions, and 2) trajectories generally originating at lower latitudes, with a larger fraction of transport from Scandinavia. These results suggest that SNAO-driven variability in atmospheric transport could contribute to variability in burning signals in the Lomonosovfonna record.

SNAO variability has been reconstructed for the past 500 years using historical documents and tree rings (Linderholm et al., 2008, 2009, 2013; Luterbacher et al., 2001). The SNAO record is primarily negative over the past 500 years, with brief positive excursions until the start of the 20th century when it shifted into its positive phase (Linderholm et al., 2009). The long-term trends in the SNAO and NAO reconstructions are generally similar, and there are some common features on centennial time scales (Fig. 8; Trouet et al., 2009). Both NAO and SNAO records exhibit a positive excursion from 1500-1650 CE, during

a period of elevated p-HBA in Lomonosovfonna. After 1400 CE, VA remains low and does not show similar variability to p-HBA. This incoherence between the records could be due to the change in atmospheric circulation patterns after 1400 CE when the Svalbard and Siberia ice core records diverge.

## 3.8    Lomonosovfonna ice core VA/p-HBA ratios

The mean VA/p-HBA ratio for the Lomonosovfonna ice core using the 10-year bin averages of each record is 0.40±0.25

(n=79). Both compounds are produced during the smoldering phase of burning (Akagi et al., 2011; Legrand et al., 2016; Simoneit, 2002) and both are produced from combustion of major boreal forest tree species. Short term changes in the ratio most likely reflect the changing contributions from various source regions with different ecosystems. Longer term changes in the ratio could reflect changes in ecology/biogeography (i.e. shifts between conifer and broadleaf forests or grasslands) or changes in atmospheric transport patterns. As noted earlier, analysis of back trajectories suggests that boreal forests are the

principle source regions for this ice core, with minor contributions from tundra and temperate forests.

     The range of VA/p-HBA ratios observed in the Lomonosovfonna ice core is consistent with the laboratory combustion studies of boreal forest tree species. Combustion studies have been conducted on several conifers characteristic of North American and European boreal forests. North American conifer (Lodgepole pine, Sitka spruce, Douglas fir, and Mountain hemlock) combustion yielded VA/p-HBA weight ratios ranging from 0.40-0.99 (Oros and Simoneit, 2001a) (Table S2, Supplemental Material).

Specific North American conifers produce only one of the two compounds. For example, Eastern White pine produced only VA, and Noble fir produced only p-HBA (Oros and Simoneit, 2001a). European conifers and peat burning produced VA/p-HBA weight ratios ranging from 0.07-8.75 (Iinuma et al., 2007). Combustion of a German peat sample yielded a low VA/p-HBA ratio of 0.08 (Iinuma et al., 2007; Oros et al., 2006). Combustion of a tundra grass sample from the Canadian Yukon Territories produced p-HBA only, but at rates 1000-fold less than conifers (Oros et al., 2006). Deciduous tree species produced roughly

1000-fold more VA than conifers (mg VA/kg fuel burned), and deciduous species did not produce detectable levels of p-HBA (Oros and Simoneit, 2001a, b). Thus even a small fraction of air mass trajectories from temperate forests could influence the VA/p-HBA ratio.

     We are not aware of laboratory combustion studies of the actual species comprising Siberian forests or tundra. This is a major gap in the knowledge base needed to interpret Arctic ice core data. Similarly, very few studies to date have reported

VA/p-HBA ratios for ambient Arctic aerosols. Fu et al. (2009) reported ratios ranging from 0.16-2.2 in weekly aerosol samples collected at Alert, covering the range observed in the ice core.

There are significant long-term changes in the Lomonosovfonna VA/p-HBA ratio over time. The ratio is relatively high during the MCA (0.8), decreases by a factor of two from 1200-1400 CE, remains low through the LIA until around 1800 CE (Fig. 3). There is also an increase in VA/p-HBA after 1800, although VA is close to the detection limit and the uncertainty in the ratio is consequently large. Interestingly, the changes in the VA/p-HBA ratio broadly mirror changes in the phase of the paleoreconstructions of the NAO and SNAO (Fig. 8). One might speculate that the associated changes in climate and transport mentioned earlier contribute to the variations in the VA/p-HBA ratio but the specific causes are not understood at this time.

There are several multi-decadal excursions in the VA/p-HBA ratio. Ratios greater than 1 occur in the VA peaks around 1270 and 1370 CE. The second of these peaks is a notable increase in VA, with no corresponding peak in p-HBA. Conversely, around 1540 and 1620 CE, there are p-HBA peaks without a corresponding peak in the VA record. These events are probably too short to represent ecological changes, but too long to represent single fire events or seasons. Such events are worthy of further investigation.

## 4   Conclusions

The Lomonosovfonna ice core record shows that the pyrogenic aromatic acids, vanillic acid and para-hydroxybenzoic acid, are present in Arctic ice and preserved on millennial time scales. The observed temporal variability of these signals should contain information about the history of high latitude burning in Northern Europe and Siberia. VA and p-HBA are both elevated from 1200-1400 CE and decline until the Little Ice Age. Paleoclimate proxy records indicate that this transition coincides with a shift in the North Atlantic Oscillation from positive to a more negative state, but the causal basis for a relationship is not established.

On centennial and shorter timescales, the two acids exhibit some notable differences. For example, the two largest peaks in the p-HBA record around 1600 CE are not present in the VA record. Conversely, elevated levels of VA from 2000-2008 CE are not present in the p-HBA record. Such anomalies are intriguing in that they suggest significant changes in either burning patterns or atmospheric transport. Further studies of the variability of these compounds in ice cores covering the instrumental and satellite eras should be conducted.

The two millennial scale ice cores analysed for these compounds to date, Lomonosovfonna and Akademii Nauk, show intriguing similarity between the Svalbard and Siberian records at the end of the Medieval Climate Anomaly (1200-1400 CE) but dramatic differences for most of the past millennium. Such differences are surprising given that air mass trajectories based on reanalysis data indicate considerable overlap in source areas for the two ice cores. It seems that there must exist a large-scale dynamical explanation of the regional/temporal trends in these and other proxy fire records. Developing a unified interpretation of these signals will require further work.

# 5 Data availability

The data reported in this manuscript will be submitted to the NSF Artic Data Center (http://arcticdata.io/) before publication.

*Author contributions.* Mackenzie Grieman, Murat Aydin, and Eric Saltzman developed the analytical method. Mackenzie Grieman combined and measured the ice core melt samples used in this study and processed the data. Mackenzie Grieman and Eric Saltzman wrote the
5  manuscript. Margit Schwikowski and her team drilled and processed the ice core, provided samples, developed the depth-age scale, and provided comments on the manuscript. Elisabeth Isaksson initiated the Lomonosovfonna ice core drilling project, was responsible for field logistics, and provided comments on the manuscript. Murat Aydin provided comments on the manuscript.

*Competing interests.* The authors declare that they have no conflicts of interest.

*Acknowledgements.* NCEP/NCAR reanalysis data were accessed from: ftp://arlftp.arlhq.noaa.gov/pub/archives/reanalysis. We would like to
10  acknowledge S. Brütsch and M. Sigl for assistance with collecting ice core melt samples, C. McCormick for help with instrument maintenance, and T. Sutterley and A. Payne for help with coding. We would also like to acknowledge research support by a generous donation from the Jenkins Family to the Department of Earth System Science, University of California, Irvine. Funding was also provided by the National Science Foundation (ANT-0839122; PLR-1142517) and by the NSF Independent Research/Development program. Wavelet software was provided by C. Torrence and G. Compo (http://paos.colorado.edu/research/wavelets/; Torrence and Compo, 1998). The Global Charcoal
15  Database was accessed at: http://gpwg.paleofire.org.

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

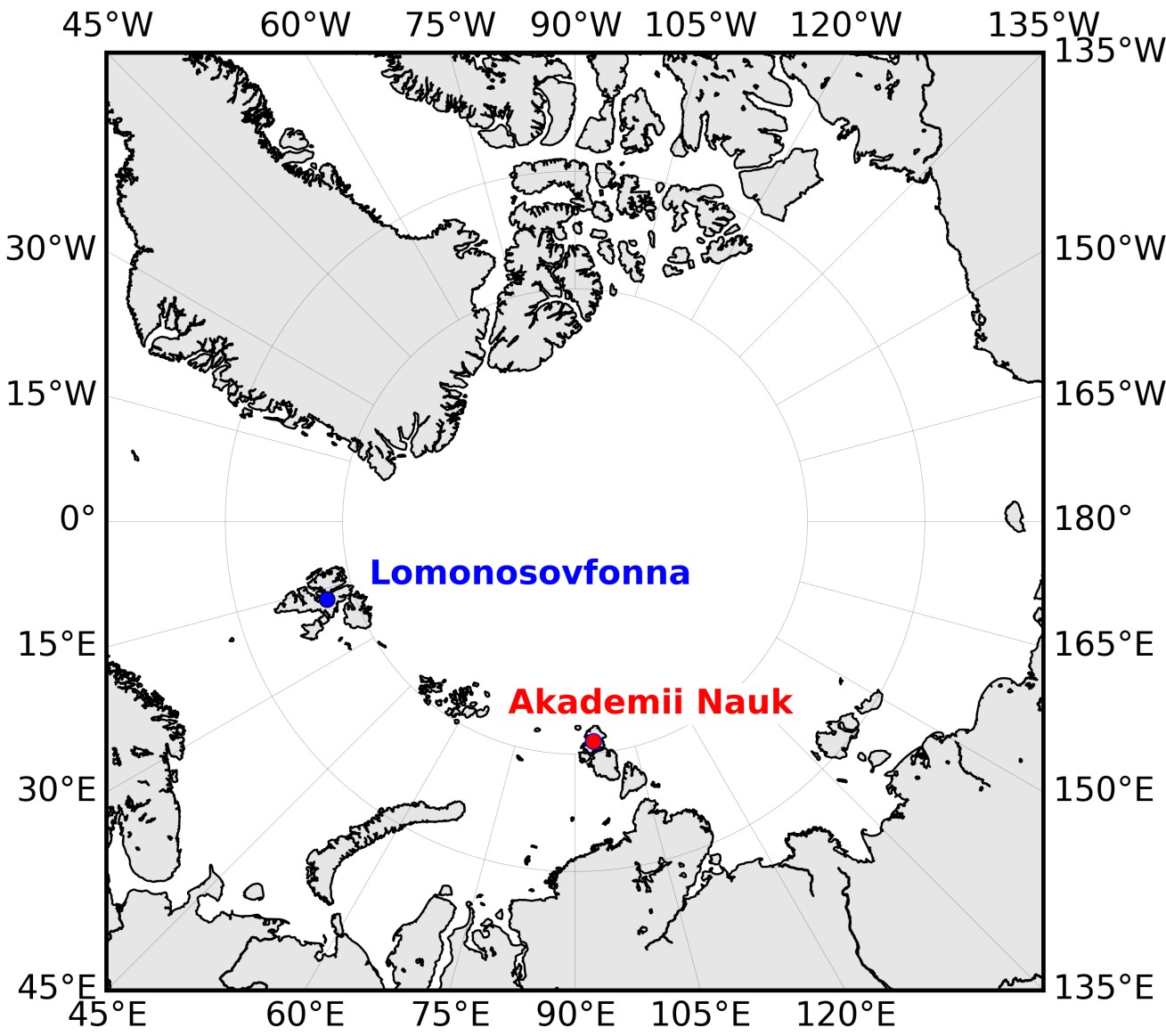

**Figure 1.** Location of Lomonosovfonna ice core drilling site on the island of Spitsbergen in Svalbard (78°49'24.4"N, 17°25'59.2"E) and the Akademii Nauk ice core drilling site on Severnaya Zemlya (80°31'N, 94°49'E). The map was produced in the Python "matplotlib" graphics environment (Hunter, 2007).

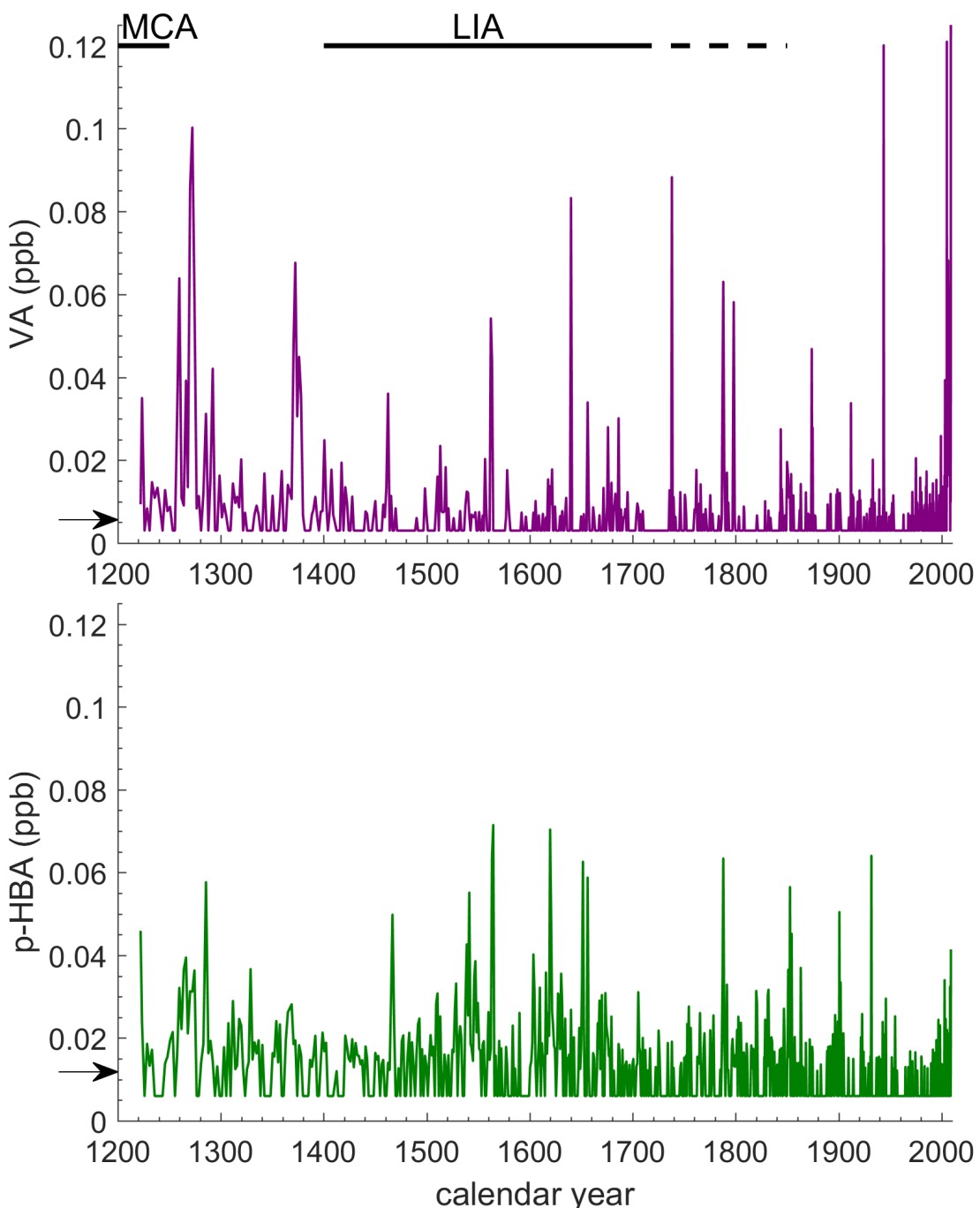

**Figure 2.** Aromatic acids in the Lomonosovfonna, Svalbard ice core. Top: vanillic acid, Bottom: p-hydroxybenzoic acid. Arrows are the detection limits. The black horizontal lines are the Medieval Climate Anomaly (MCA) and the Little Ice Age (LIA) (Mann et al., 2009). The dashed horizontal line is the extended LIA in the Svalbard region (Divine et al., 2011).

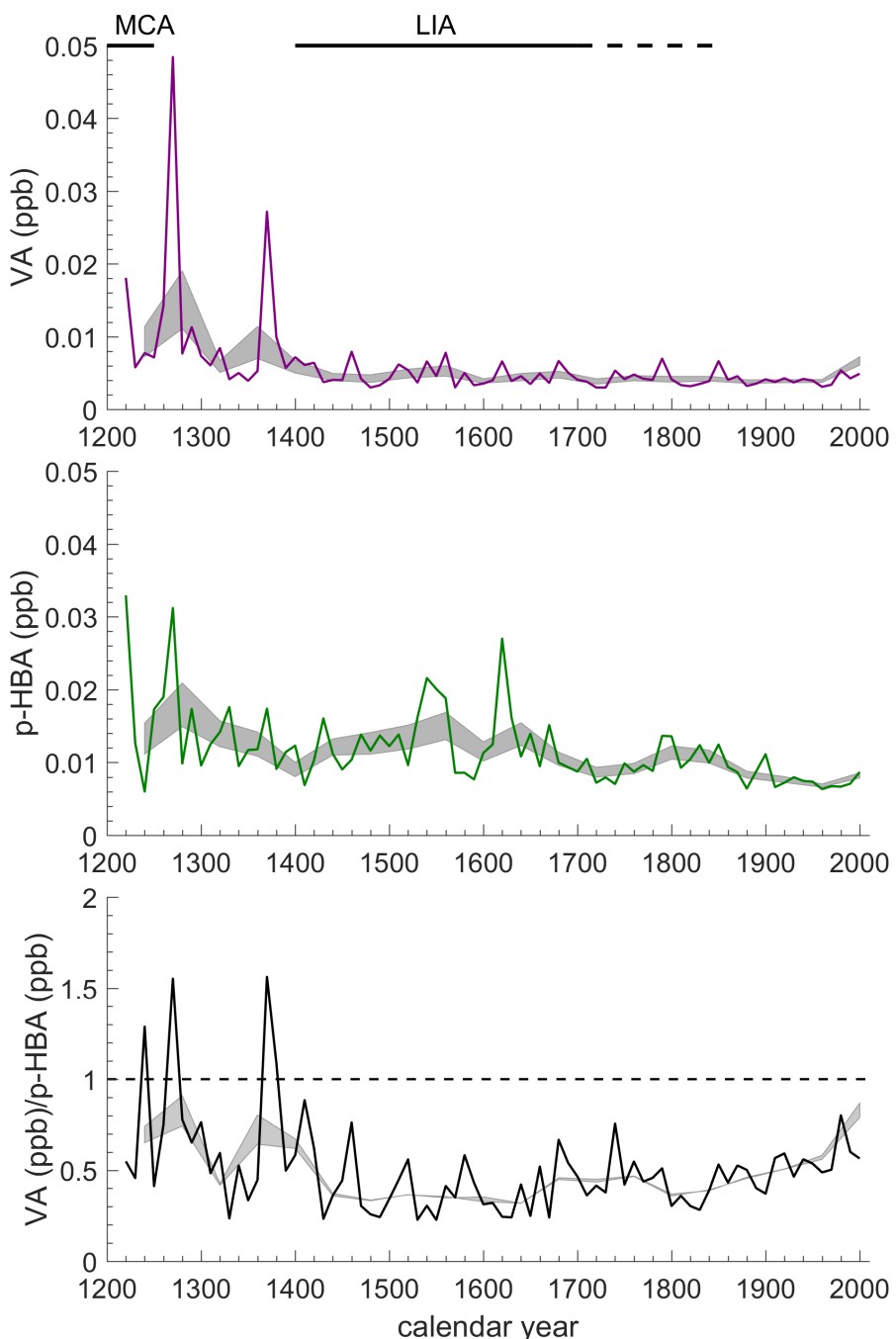

**Figure 3.** Lomonosovfonna ice core records of vanillic acid (top) and p-hydroxybenzoic acid (middle) and the ratio of vanillic acid/p-hydroxybenzoic acid (bottom). For all plots: solid lines are 10-year bin averages, gray shaded areas are 40-year bin averages of ±1 standard error. The dashed line on the bottom plot indicates a ratio of 1. The black horizontal lines are the Medieval Climate Anomaly (MCA) and the Little Ice Age (LIA) (Mann et al., 2009). The dashed horizontal line at the top is the extended LIA in the Svalbard region (Divine et al., 2011).

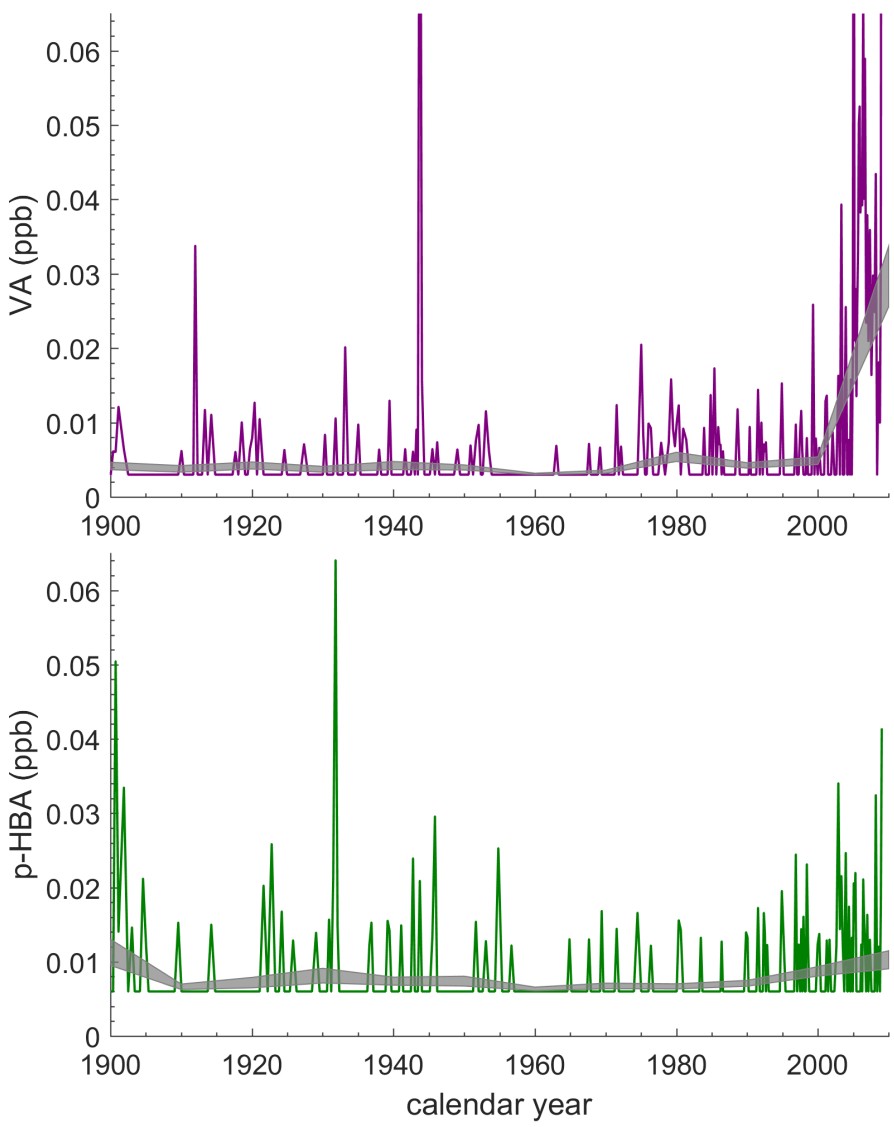

**Figure 4.** Lomonosovfonna ice core records of vanillic acid (top) and p-hydroxybenzoic acid (bottom) for the 20th century. Gray shaded areas are 10-year bin averages with ±1 standard error.

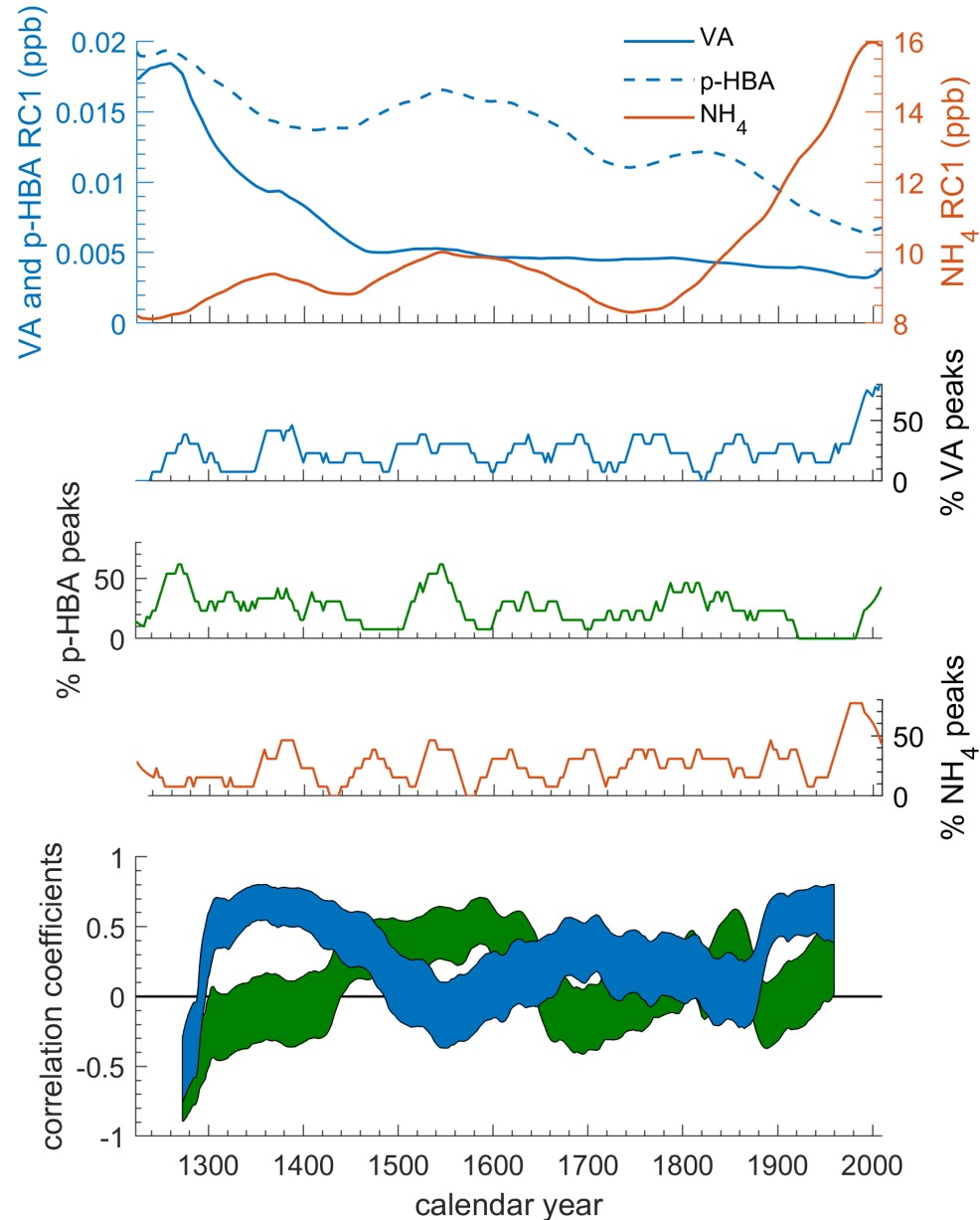

**Figure 5.** Relationships between Lomonosovfonna vanillic acid (VA), p-hydroxybenzoic acid (p-HBA), and ammonium (NH₄) using 3-year bin averaged data. 1) First component from the singular spectrum analysis of VA (blue solid line), p-HBA (blue dashed line), and NH₄ (orange line) (PC1) reconconstructed into concentration units (RC1, ppb), 2-4) Frequency of peaks in the ice core signals reconstructed using singular spectrum components 2-30 and peak threshold of 75th percentile, smoothed with a 40-year running window. 5) Correlation coefficients for the ice core peak frequencies using a 200-year running window (p<0.001). Bands are the 95% confidence intervals of the correlation coefficients of VA and ammonium (blue) and p-HBA and ammonium (green).

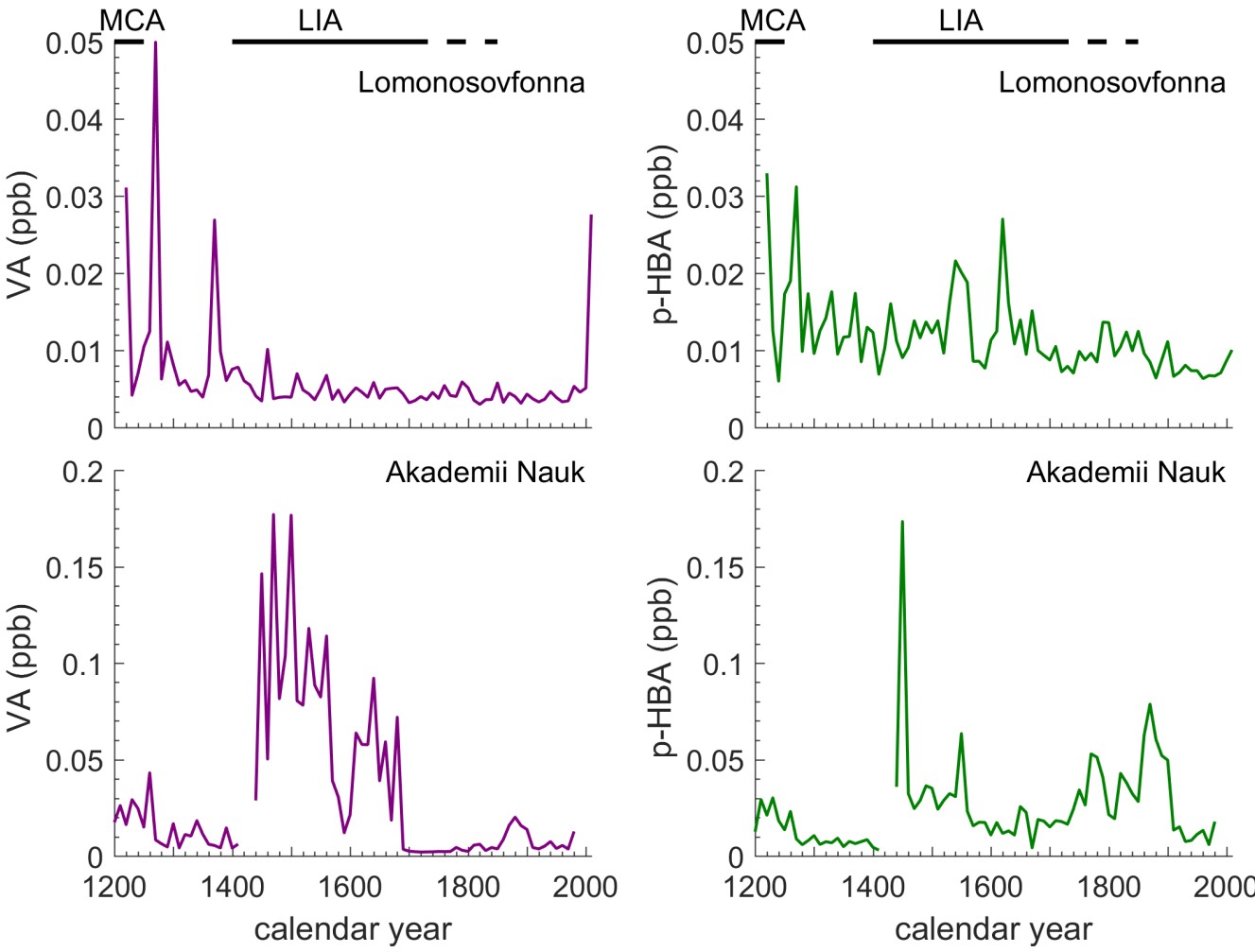

**Figure 6.** Aromatic acids in the Lomonosovfonna, Svalbard and Akademii Nauk ice cores. Left: vanillic acid, Right: p-hydroxybenzoic acid. Violet lines are 10-year bin averages of the the Lomonosovfonna ice core measurements. Green lines are the 10-year bin averages of the Akademii Nauk ice core measurements (Grieman et al., 2017). The black horizontal lines are the Medieval Climate Anomaly (MCA) and the Little Ice Age (LIA) (Mann et al., 2009). The dashed horizontal line is the extended LIA in the Svalbard region (Divine et al., 2011).

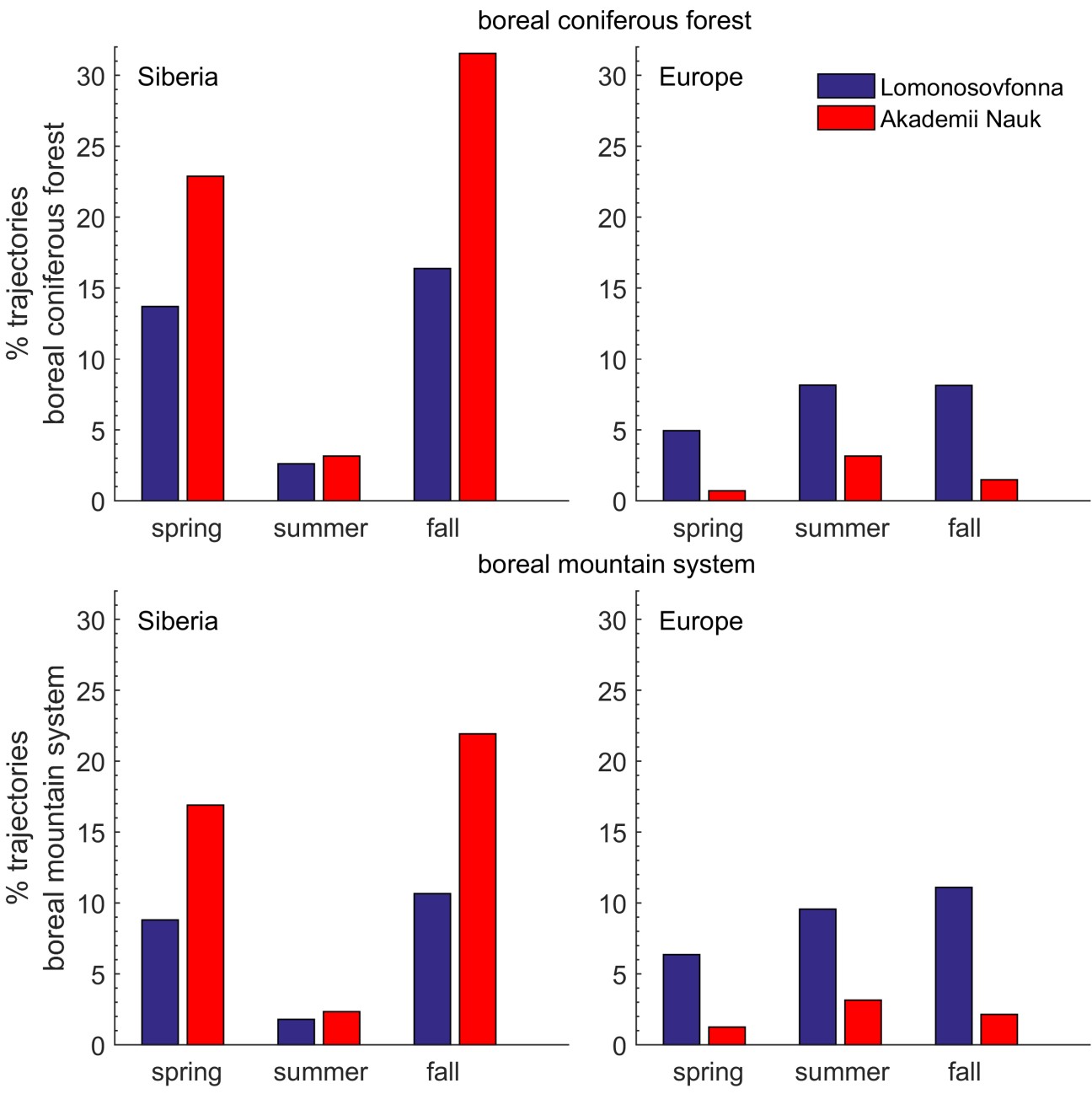

**Figure 7.** 10-day back trajectories from 2006-2015 reaching the boreal ecosystems starting from the Lomonosovfonna and Akademii Nauk ice core locations. Blue is Lomonosovfonna. Red is Akademii Nauk. Trajectories reaching: Siberia (left), Europe (right), boreal coniferous forest (top), and boreal mountain system (bottom).

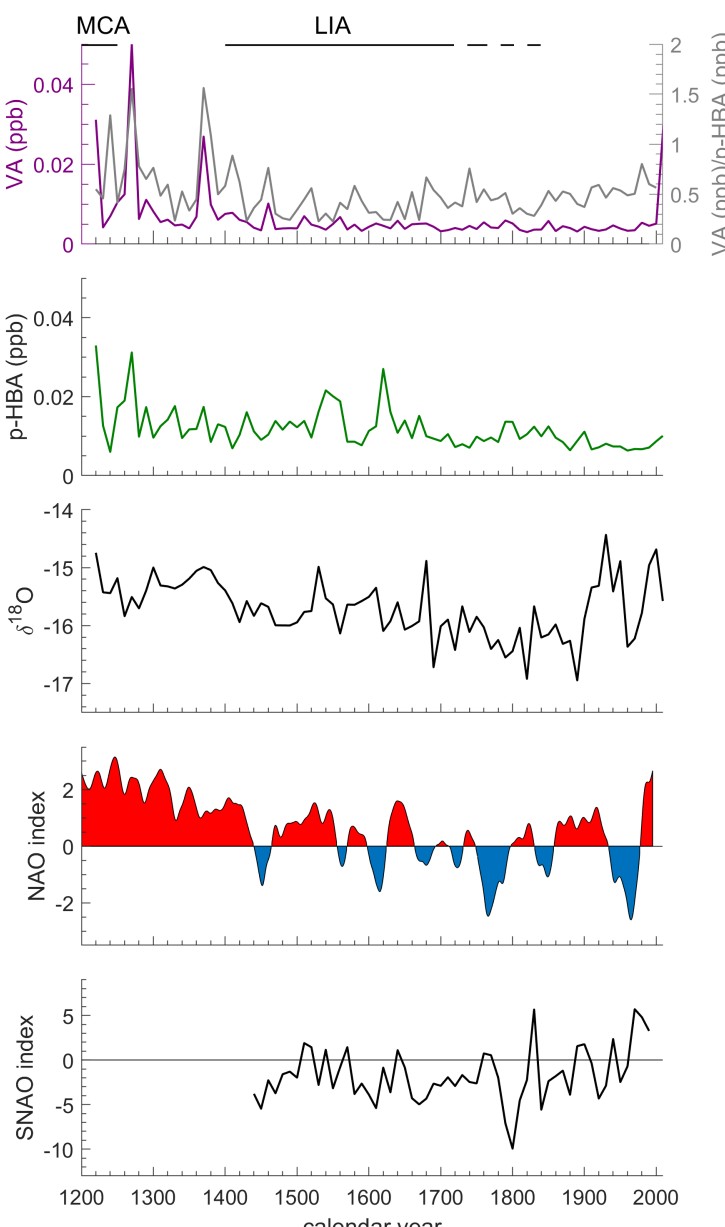

**Figure 8.** Comparison of the timing of aromatic acid signals in the Lomonosovfonna ice core over the past 800 years compared to other climate-related proxy records. From top: 10-year bin averages of Lomonosovfonna vanillic acid (violet line), ratio of vanillic acid/p-hydroxybenzoic acid (gray line), and p-hydroxybenxoic acid; 10-year bin averages of the oxygen isotope record from the Lomonosovfonna ice core (Wendl et al., 2015); North Atlantic Oscillation (NAO) index (red > 0; blue < 0; Trouet et al., 2009); and 10-year bin averages of the summer North Atlantic Oscillation (SNAO) index (Linderholm et al., 2009). The black horizontal lines are the timing of the Medieval Climate Anomaly (MCA) and the Little Ice Age (LIA) (Mann et al., 2009). The dashed horizontal line is the extended LIA in the Svalbard region (Divine et al., 2011).

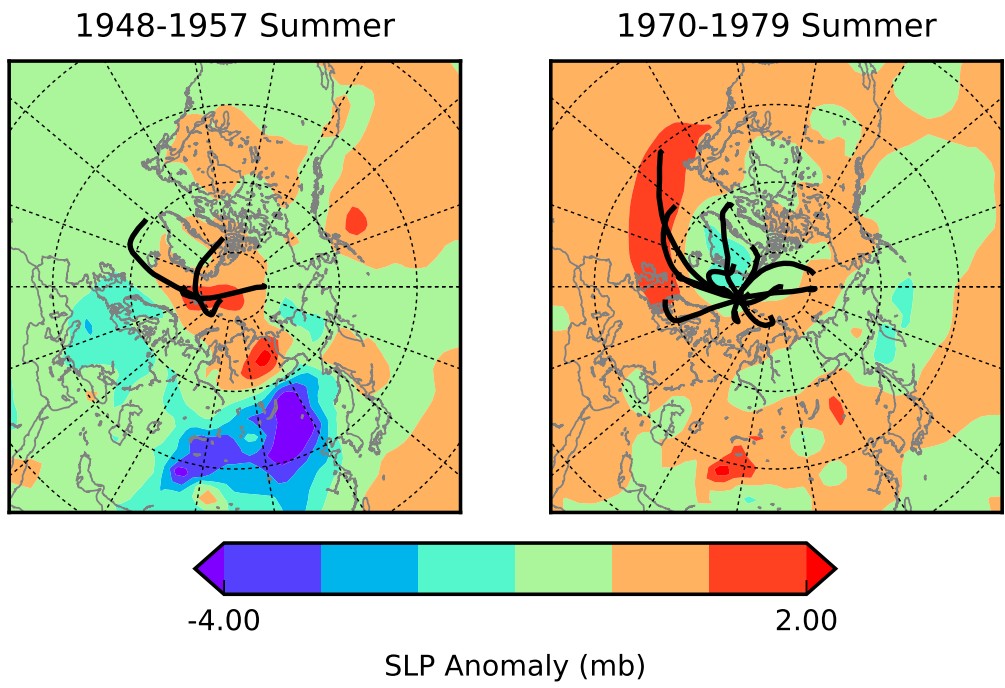

**Figure 9.** 10-day clustered back trajectories starting from the Lomonosovfonna ice core location superimposed on sea level pressure anomalies for summer (June-August) of 1948-1957 (negative SNAO) and 1970-1979 (positive SNAO). Anomalies are relative to the 1948-2017 mean of NCEP/NCAR reanalysis data.