# Peer review of "Aromatic acids in an Arctic ice core from Svalbard: a proxy record of biomass burning"

_Climate of the Past, 2017_

## Referee Comment (RC1) · Anonymous Referee #1 · 3 Nov 2017

General Comments

This work proposes very interesting records of two aromatic acids in an Arctic ice core on millennial time scales. The comparison between Svalbard and Siberian ice cores of these two compounds underlines dramatic differences for the most of the past millennium, underlining the extreme need to have several records to better define the past atmospheric circulation and post-depositional processes. The investigation about the past changes of fire regimes can help to better understand how climate change may influence fire and its impact on the carbon cycle in the future. I saw that you have evaluated the charcoal records on the supplementary information, but I think that this

part should be reported in the main manuscript with other considerations. I suggest to insert a new paragraph in the main manuscript with the comparison with other biomass burning proxy records. For example, Rubino et al (2015) reported that ammonium and nss -K can be used as biomass burning proxies, and these data are available in the same ice core (Wendl, ACP, 2015). Some authors of this paper have collaborated to publish the paper of aromatic acids in the Akademii Nauk ice core, in which a good comparison with other proxies (for example with levoglucosan) is reported. In literature, some authors (for example Zangrando et al., 2016) suggested that vanillic acid can have other sources beyond biomass burning. I think that this new paragraph could enhance the reliability of the proposed proxies. In the manuscript (and also in the title), the authors consider p-hydroxybenzoic acid as a methoxy aromatic acid but p-HBA does not have the methoxy moiety. Please check and correct. The paper is generally well-written and clear, but I have suggested some corrections. Therefore, I recommend this paper for publication with major revisions.

Specific comments

Introduction. Page 2. Lines 1-10. You reported only a list of possible biomass burning tracers, but I suggested to better describe the advantages and the disadvantages of each marker. I recommend to improve this part of the introduction.

Specific comments Title. The use of "methoxy" in the title is wrong because p-hydroxybenzoic acid is not a methoxy phenol. I suggest to remove "methoxy".

Page 4. Line 32. You detected VA using two different transitions (167>108 and 167>152) while p-HBA with only one transition (137>93). The quantitative method using HPLC-MS/MS or IC-MS/MS requires the monitoring of two transitions where the most intense transition was used to quantify the compound and the other one was used to confirm the identity of compound.

Page. 5. Line 5. Have you evaluated the contamination during the proceeding? Have you subtract the blank values?

Page 5. Line 8. You reported that you analyzed 993 samples, but in page 4-line 26 you wrote that you had 997 samples. Please correct this discrepancy.

Page 7. Lines 25-31 and figure 7. Why you have reported two different NAO indexes from two different references? Which is the difference between two records?

Page 10. Paragraph 3.6. In this paragraph you described the behavior of proxies and their possible modification occurred due to atmosphere/snow interactions. I think that the discussion about "potential for post depositional modification of VA and p- HBA" should be reported before of "Relationship to atmospheric circulation and climate".

Technical comments

Abstract. Line 5. Please correct "1,000 ng/l" with "1,000 ng L-1) Page 5. Line 10. Please add "limit" after "detection". Page 5. Line 10. Please add "0" before of ".006". Figures 6 and 8. Please can you specify the period that you consider to calculate the back trajectories.

---

## Referee Comment (RC2) · L. Michel (Referee) · 5 Nov 2017

Comments by Michel LEGRAND on the manuscript entitled "Methoxy aromatic acids in an Arctic ice core from Svalbard: a proxy record of biomass burning" by Mackenzie M. Grieman et al.

This paper reports on the concentrations of aromatic acids (vanillic and parahydroxybenzoic) measured along a Svalbard Arctic ice core covering more than 800 years. These acids that are used as proxies of biomass burning were measured by using a technique (IC-ESI-MS/MS) recently developed at the department of Earth System Science at Irvine (CA). The obtained records are then compared to those derived from

another ice core extracted at the Akademii Nauk ice cap located in the Eurasian Arctic. Data on past frequency of boreal fires are of great importance since the boreal forest represents an important carbon reservoir and experiences predominantly natural fires of which the severity is expected to change with future warming and the subsequent modification of spring/summer/fall conditions. In contrast to Canadian fires, Siberian fires are far less documented except for the very last decades when satellite data has strongly increased the accuracy of estimated burned area of this region. The Svalbard experiencing air masses from Siberia and to a lesser extent from Europe, this paper provides new information together with those recently extracted from the Akademii Nauk on Siberian fires over the past. The paper is therefore of great interest for scientific communities working on forest fire records in ice cores and lake sediments as well as for the general topic of climate/fire conditions/vegetation interactions. Overall the manuscript is well organized and clearly written. The discussion of data is generally well conducted and very good. I therefore recommend publication of the manuscript, after authors consider the following (minor) points rise below.

Overall comment: Several recent papers have launched the discussion on the quality of different potential proxies to reconstruct past biomass burning activity from ice (see Rubino et al., 2016; Legrand et al., 2016). There are two types of fire proxies: (1) minor organic species like levoglucosan and different resinic and aromatic acids, and (2) various major inorganic species including ammonium that was extensively used to reconstruct North America fires in Greenland ice (see Savarino and Legrand, 1998; Fischer et al., 2015, for instance). There are now two studies having investigated past biomass burning from sites exposed to air masses coming from Eurasia (Grieman et al., 2016, and the present work) by using organic markers. At least for the Svalbard ice core, major ions including ammonium are also available (Wendl et al., 2015). Checking the ammonium profile (Figures 3 and 4 in Wendl et al.), I see three time periods with elevated ammonium levels (around 1370, 1545, and 1900) but nothing in 1300. Can we conclude from that ammonium is not an adequate biomass burning tracer in this region? Is this difference for ammonium between Arctic and Greenland sites related

to difference of altitude of plumes (more scavenging at the low elevated marine site of Svalbard???). Please comment if possible. Please also note that Wendl et al., discussed the ammonium record in Svalbard as follows: "A period of exceptional high fire activity around 1600–1680 in Siberian boreal forests of Eurasia was detected in the ice core fire tracer records from the Siberian Altai (Eichler et al., 2011). This unique period did not lead to a maximum in the Lomo09 NH4+ record. Therefore, we conclude that biomass burning is not a major source of NH4+ arriving at Svalbard." Fischer, H., Schüpbach, S., Gfeller, G., Bigler, M., Röthlisberger, R., Erhardt, T., Stocker, T. F., Mulvaney, R., and Wolff, E. W.: Millennial changes in North American wildfire and soil activity over the last glacial cycle, Nat. Geosci., 8, 723–727, doi:10.1038/NGEO2495, 2015. Eichler, A., Tinner, W., Brusch, S., Olivier, S., Papina, T., and Schwikowski, M.: An ice-core based history of Siberian forest fires since AD 1250, Quaternary Sci. Rev., 30, 1027–1034, 2011. Savarino, J. and Legrand, M.: High northern latitude forest fires and vegetation emissions over the last millennium inferred from the chemistry of a central Greenland ice core, J. Geophys. Res. Atmos., 103, 8267–8279, 1998.

Minor points: Abstract: Please specify for which season air mass back trajectories were computed and for how many days. Page 4, Line 5-8: Please specify for how many days air mass back trajectories were computed. Page 2, line 9: Please clarify the reference Rubino et al. (2015): In my record the paper had appeared in 2016: The Anthropocene Review 2016, Vol. 3(2) 140–162. Page 6, Line 30-33: Please specify for how many days air mass back trajectories were computed for both sites (5 days ?, 10 days ?). Figure 7: Sodium at GISP 2: This figure will not really convince the reader that the NAO influences the sodium record in central Greenland. By the way, what tell us the sodium record at the Svalbard site (available in Wendl et al., 2015) in Figure 3 and 4. May be a comment is welcome in the discussion here (in section 3.4).

End of the review

---

## Referee Comment (RC3) · Anonymous Referee #3 · 21 Nov 2017

The location of the Lomonosovfonna ice core, in conjunction with the author's previous work on the Akademii Nauk ice core provide the opportunity to examine biomass burning across the high Arctic. The authors use similar techniques for both ice cores, where the biomass burning records are similar for approximately 200 years, but then deviate after 1400 CE until the present. The authors ascribe this divergence to an "atmospheric reorganization" due to possible changes in the SNAO, but this argument can be more thoroughly developed, as detailed in the points below. While few easily-accessed paleoclimate records exist near the Akademii Nauk ice core, Svalbardhas ben extensively studied for decades. It is surprising that the authors only briefly touch on the extensive paleoclimate information and modern aerosol transport datafrom Svalbard in their

study.

Page 1 Lines 5 and 6: "Vanillic acid levels are high (below the limit of detection to 0.1 ppb) from 1200-1400 CE, then gradually decline into the 20th century." Concentrations below the level of detection cannot be high by definition.

Page 1 Line 9 to 10: Are Siberia and Europe the primary source regions throughout the time period of the entire study? Or are they the primary modern source regions?

Lines 19 to 20: "Boreal wildfire areal extent appears to have increase significantly with warming during the past few decades" needs a citation.

Page 3 Lines 20-29: Why do you use ten-year bin averages rather than, for example, 10-year moving averages? If the dating uncertainty below 80 m is 10 years, then are ten-year bin averages too narrow of a time frame? It is essential to explain your reasoning in this section.

Page 4 Line 5: Why did you choose to start the trajectories at 100 m above the ice surface?

Page 4 Lines 8 and 9: What is the latitudinal boundary for North America, Siberia and Europe in your study?

Section 3.1: Either refer to this section in Page 3 Lines 20-29 or else move the entire section to immediately follow the current Page 3 lines 20-29 where you first describe that you use 10-year bins for your data.

Page 5 Lines 13 and 14: Were the geometric means and standard deviations used because of the skewness? I think that this is what you would like to say, but please rephrase to your meaning is clear.

Page 5 Line 15: At the deepest section of the core, how many samples do you have in each 10-year bin?

Page 5 Lines 28-30: Can this long-term decreasing trend be caused by decomposition

or degradation of VA and p-HBA through time? (Refer the reader to Section 3.6 to demonstrate that you have considered these possibilities).

Section 3.4 and Conclusions: Stating that an "atmospheric reorganization" due to changes in the SNAO affects the differences in biomass burning tracers is quite a bold statement. Although you describe the spatial patterns of the SAO, a figure can better demonstrate the influence of the SNAO on transport affecting these two ice cores sites. The back trajectories for the positive SNAO index (1970-1979 CE) and negative (1948-1957 CE) can help depict the source regions and transport paths. Figure 2 of Folland et a., (2009) is an excellent example of the spatial extent of the summer NAO. However, plotting an example of the spatial patterns for a positive SNAO (1970-1979 CE) and negative SNAO index (1948-1957 CE) can also add essential support to your argument. Section 3.5: You mention that "the long-term trends in the VA/p-HBA ratio presumably reflect changes in the relative contributions of fuel types or changes in atmospheric transport". I kept expecting you to tie these possible changes in atmospheric transport back to the discussion of the SNAO. This lack of a mention of causes of these atmospheric changes is surprising in light of the previous section.

---

## Author Comment (AC1) · 30 Jan 2018

The referee raised several good points and we appreciate the comments. The manuscript has been modified as described below to take them into account. Referee comments are numbered and our responses follow.

1. I saw that you have evaluated the charcoal records on the supplementary information, but I think that this part should be reported in the main manuscript with other considerations.

As recommended, the comparison to charcoal records has been moved into the Results and Discussion section of the main manuscript as Section 3.5.

2. I suggest to insert a new paragraph in the main manuscript with the comparison with other biomass burning proxy records. For example, Rubino et al (2015) reported that ammonium and nss -K can be used as biomass burning proxies, and these data are available in the same ice core (Wendl, ACP, 2015). Some authors of this paper have collaborated to publish the paper of aromatic acids in the Akademii Nauk ice core, in which a good comparison with other proxies (for example with levoglucosan) is reported. Introduction. Page 2. Lines 1-10. You reported only a list of possible biomass burning tracers, but I suggested to better describe the advantages and the disadvantages of each marker. I recommend to improve this part of the introduction.

The text in the introduction has been expanded as suggested.

3. In the manuscript (and also in the title), the authors consider p-hydroxybenzoic acid as a methoxy aromatic acid but p-HBA does not have the methoxy moiety. Please check and correct. Specific comments Title. The use of "methoxy" in the title is wrong because phydroxybenzoic acid is not a methoxy phenol. I suggest to remove "methoxy".

We greatly appreciate the reviewer catching this obvious (and embarrassing) mistake! "Methoxy" is now removed from the title and the rest of the manuscript.

4. Page 4. Line 32. You detected VA using two different transitions (167>108 and 167>152) while p-HBA with only one transition (137>93). The quantitative method using HPLC-MS/MS or IC-MS/MS requires the monitoring of two transitions where the most intense transition was used to quantify the compound and the other one was used to confirm the identity of compound.

Unfortunately, p-HBA has only one mass transition representing a significant fraction of the total signal (137→93). It was therefore not possible to confirm identity using another mass transition. We have seen no evidence in terms of peak shape or retention time

to cast doubt on the identity of the peak.

5. Page. 5. Line 5. Have you evaluated the contamination during the proceeding? Have you subtract the blank values?

For this study, full procedural blanks covering field collection and sample melting were not available. We did routinely analyze laboratory blanks along with samples. These did not exhibit detectable peaks at the mass transitions for VA or p-HBA. During this and previous studies, we have not experienced contamination of these compounds at significant levels.

As noted in the Methods section, the limits of detection were calculated using 3x the standard deviations of the blank. Measurements at or below the detection limit were reported as $\frac{1}{2}$ of the limits of detection.

6. Page 5. Line 8. You reported that you analyzed 993 samples, but in page 4-line 26 you wrote that you had 997 samples. Please correct this discrepancy.

We collected 997 samples but analyzed only 993. The text has been modified to refer only to the number analyzed.

7. Page 7. Lines 25-31 and figure 7. Why you have reported two different NAO indexes from two different references? Which is the difference between two records?

We agree that the two NAO records were similar and removed the shorter of the two records.

8. Page 10. Paragraph 3.6. In this paragraph you described the behavior of proxies and their possible modification occurred due to atmosphere/snow interactions. I think that the discussion about "potential for post depositional modification of VA and p- HBA" should be reported before of "Relationship to atmospheric circulation and climate".

Agreed. The discussion of postdepositional modification is now Section 3.3.

9. Abstract. Line 5. Please correct "1,000 ng/l" with "1,000 ng L-1)

Done

10. Page 5. Line 10. Please add "limit" after "detection".

"Below detection" was changed to "below the limits of detection."

11. Page 5. Line 10. Please add "0" before of ".006".

Done

12. Figures 6 and 8. Please can you specify the period that you consider to calculate the back trajectories.

These figures are now figures 7 and 9. The figure 7 caption was revised as follows: Figure 7. "…10-day back trajectories from 2006-2015 reaching the boreal ecosystems starting from the Lomonosovfonna and Akademii Nauk ice core locations…."

Figure 9 has been removed and replaced with a different figure.

---

## Author Comment (AC2) · 30 Jan 2018

The referee raised several points and we appreciate the comments. The manuscript has been modified as described below to take the comments into account. Referee comments are numbered and our responses follow.

1. Checking the ammonium profile (Figures 3 and 4 in Wendl et al.), I see three time periods with elevated ammonium levels (around 1370, 1545, and 1900) but nothing in 1300. Can we conclude from that ammonium is not an adequate biomass burning tracer in this region? Is this difference for ammonium between Arctic and Greenland sites related to difference of altitude of plumes (more scavenging at the low elevated

marine site of Svalbard???).

In an effort to answer this question, we attempted a quantitative examination of the co-variability of Lomonosovfonna VA, p-HBA, and ammonium. The results are reported in section 3.4.

2. Abstract: Please specify for which season air mass back trajectories were computed and for how many days.

Page 1, line 9 was changed to "Air mass back trajectories for a decade of fire seasons (March-November, 2006-2015) indicate that Siberia and Europe are the principle source regions for wildfire emissions reaching the Lomonosovfonna site."

3. Page 4, Line 5-8: Please specify for how many days air mass back trajectories were computed.

Page 4, Lines 5-8 were changed to: "The 10-day back trajectories were started at 100 m above the ice surface at 12:00 AM and 12:00 PM local time for three separate 10-year periods, 1948-1957, 1970-1979, and 2006-2015 CE."

4. Page 2, line 9: Please clarify the reference Rubino et al. (2015): In my record the paper had appeared in 2016:

Corrected.

5. Page 6, Line 30-33: Please specify for how many days air mass back trajectories were computed for both sites (5 days ?, 10 days ?).

Page 6, Lines 30-33 were changed to "10-day back trajectories were computed for the Akademii Nauk site using the same methods as those described above from 2006-2015 CE (section 2.2; Grieman et al., 2017). The 10-day back trajectories show that both Lomonosovfonna and Akademii Nauk sites are influenced by air masses transecting Eurasian forested regions (Fig. 7; Table S1).

6. Figure 7: Sodium at GISP 2: This figure will not really convince the reader that the

NAO influences the sodium record in central Greenland.

These data did not add much to the discussion and have been removed from figure 8.

7. By the way, what tell us the sodium record at the Svalbard site (available in Wendl et al., 2015) in Figure 3 and 4.

The sodium record for this core published by Wendl et al. (2015) shows a long term declining trend, with centennial variability that is generally similar in character to VA, p-HBA and other parameters in this core. The Wendl et al. (2015) paper provided little discussion about the causes of the sodium variability. Presumably, the seasalt record reflects changes in the frequency of air mass trajectories from the North Atlantic, as well as the intensity of storms and one might speculate that both could be related to changes in the phase of the NAO. A serious analysis of seasalt data covering the period of satellite era would be worthwhile, but outside the scope of this paper.

———————————————————

---

## Author Comment (AC3) · 30 Jan 2018

The referee raised several points and the comments are appreciated. The manuscript has been changed as described below to take them into account. Referee comments are numbered and our responses follow.

1. Page 1 Lines 5 and 6: "Vanillic acid levels are high (below the limit of detection to 0.1 ppb) from 1200-1400 CE, then gradually decline into the 20th century." Concentrations below the level of detection cannot be high by definition.

This sentence has been changed to: "Vanillic acid levels are high (maximum of 0.1

ppb) from 1200-1400 CE, then gradually decline into the 20th century."

2. Page 1 Line 9 to 10: Are Siberia and Europe the primary source regions throughout the time period of the entire study? Or are they the primary modern source regions?

This sentence has been changed to: "10-day air mass back trajectories for a decade of fire seasons (March-November, 2006-2015) indicate that Siberia and Europe are the principle modern source regions for wildfire emissions reaching the Lomonosovfonna site."

3. Lines 19 to 20: "Boreal wildfire areal extent appears to have increase significantly with warming during the past few decades" needs a citation.

This sentence has been removed.

4. Page 3 Lines 20-29: Why do you use ten-year bin averages rather than, for example, 10-year moving averages? If the dating uncertainty below 80 m is 10 years, then are ten-year bin averages too narrow of a time frame? It is essential to explain your reasoning in this section.

The choice of bin-averaging over moving averages is not significant. To demonstrate this, we added moving averages to Figure S8 for comparison to the bin average. The differences between the two are very slight and would not impact the interpretation.

We note that the dating uncertainty is an issue of absolute age assignment, not time resolution, so it is not really related to the issue of resolving signals.

5. Page 4 Line 5: Why did you choose to start the trajectories at 100 m above the ice surface?

We were initially concerned about how representative ground-level back trajectories would be given the potential for shallow, highly stable boundary layers, inversions, etc. So, we conducted trajectories starting at multiple levels from the surface to 500m above surface. These all gave very similar results. We arbitrarily selected 100m.

6. Page 4 Lines 8 and 9: What is the latitudinal boundary for North America, Siberia and Europe in your study?

The following sentence was added beginning on Page 4, lines 19-20: "The boundaries of North America, Siberia and Europe were defined using global self-consistent, hierarchical, high-resolution geography database GIS shapefiles (Wessel et al., 1996)."

Wessel, P., and W. H. F. Smith, A Global Self-consistent, Hierarchical, High-resolution Shoreline Database, J. Geophys. Res., 101, 8741-8743, 1996.

7. Section 3.1: Either refer to this section in Page 3 Lines 20-29 or else move the entire section to immediately follow the current Page 3 lines 20-29 where you first describe that you use 10-year bins for your data.

A reference to Section 3.1 has been added to Page 4, line 10: "Ten-year bin averages are used to illustrate short-term variability in the data (see Section 3.1)."

8. Page 5 Lines 13 and 14: Were the geometric means and standard deviations used because of the skewness? I think that this is what you would like to say, but please rephrase to your meaning is clear.

Page 5, Lines 13-14 were changed to the following: "Geometric means and standard deviations were used for all statistics because the frequency distribution of the data was skewed towards lower concentrations."

9. Page 5 Line 15: At the deepest section of the core, how many samples do you have in each 10-year bin?

The time bin from 1221-1225 contains two samples. The first 10-year time bin from 1225-1235 contains 4 samples. The long-term declining trend also is shown using 40-year bin averages.

10. Page 5 Lines 28-30: Can this long-term decreasing trend be caused by decomposition or degradation of VA and p-HBA through time? (Refer the reader to Section 3.6

to demonstrate that you have considered these possibilities).

Degradation of VA and p-HBA would cause a long-term increasing trend because VA and p-HBA deeper in the ice core would have degraded.

11. Section 3.4 and Conclusions: Stating that an "atmospheric reorganization" due to changes in the SNAO affects the differences in biomass burning tracers is quite a bold statement. Although you describe the spatial patterns of the SAO, a figure can better demonstrate the influence of the SNAO on transport affecting these two ice cores sites. The back trajectories for the positive SNAO index (1970-1979 CE) and negative (1948-1957 CE) can help depict the source regions and transport paths. Figure 2 of Folland et a., (2009) is an excellent example of the spatial extent of the summer NAO. However, plotting an example of the spatial patterns for a positive SNAO (1970-1979 CE) and negative SNAO index (1948-1957 CE) can also add essential support to your argument.

We agree that the statement was perhaps too strong. We revised Page 10, line 15 to: "We suggest that a change of high latitude northern hemisphere atmospheric circulation patterns occurred at this time,..."

We took the suggestion to make a new figure. Figure 9 was removed because it did not add to the argument. Figure 9 was replaced with a new figure. The following text has been added, Page 10, lines 30-33: "Figure 9 shows the major spatial clusters of 10-day air mass back trajectories for each time period (computed using Hysplit) superimposed on the sea level pressure (SLP) anomalies relative to mean SLP from 1948-2017. The high SNAO period is characterized by 1) high pressure over Scandinavia, favoring drier conditions, and 2) trajectories generally originating at lower latitudes, with a larger fraction of transport from Scandinavia."

12. Section 3.5: You mention that "the long-term trends in the VA/p-HBA ratio presumably reflect changes in the relative contributions of fuel types or changes in atmospheric transport". I kept expecting you to tie these possible changes in atmospheric transport

back to the discussion of the SNAO. This lack of a mention of causes of these atmospheric changes is surprising in light of the previous section.

The following has been added to section 3.5: "There are significant long-term changes in the Lomonosovfonna VA/p-HBA ratio over time. The ratio is relatively high during the MCA (0.8), decreases by a factor of two from 1200-1400 CE, remains low through the LIA until around 1800 CE (Fig. 3). There is also an increase in VA/p-HBA after 1800, although VA is close to the detection limit and the uncertainty in the ratio is consequently large. Interestingly, the changes in the VA/p-HBA ratio broadly mirror changes in the phase of the paleoreconstructions of the NAO and SNAO (Fig. 8). One might speculate that the associated changes in climate and transport mentioned earlier contribute to the variations in the VA/p-HBA ratio but the specific causes are not understood at this time."